# Personalized Cancer Medicine in the Media: Sensationalism or Realistic Reporting?

**DOI:** 10.3390/jpm11080741

**Published:** 2021-07-28

**Authors:** Katherine Hicks-Courant, Jenny Shen, Angela Stroupe, Angel Cronin, Elizabeth F. Bair, Sam E. Wing, Ernesto Sosa, Rebekah H. Nagler, Stacy W. Gray

**Affiliations:** 1Division of Gynecologic Oncology, University of Pennsylvania, Philadelphia, PA 19104, USA; katherine.hicks-courant@pennmedicine.upenn.edu; 2Department of Psychology, The State University of New York at Stony Brook, Stony Brook, NY 11794, USA; js4522@tc.columbia.edu; 3Patient Reported Outcomes, Pharmerit International, Cambridge, MA 02142, USA; astroupe@pharmerit.com; 4Corrona, LLC, Waltham, MA 02451, USA; acronin@corrona.org; 5Department of Medical Ethics and Health Policy, Perelman School of Medicine, University of Pennsylvania, Philadelphia, PA 19104, USA; efbair@pennmedicine.upenn.edu; 6Department of Population Sciences, City of Hope Comprehensive Cancer Center, Duarte, CA 91010, USA; swing@coh.org (S.E.W.); Ernesto.a.sosa@gmail.com (E.S.); 7Hubbard School of Journalism & Mass Communication, University of Minnesota, Minneapolis, MN 55455, USA; nagle026@umn.edu

**Keywords:** news media, personalized medicine, genomic testing, targeted therapies, public awareness

## Abstract

Background: Given that media coverage can shape healthcare expectations, it is essential that we understand how the media frames “personalized medicine” (PM) in oncology, and whether information about unproven technologies is widely disseminated. Methods: We conducted a content analysis of 396 news reports related to cancer and PM published between 1 January 1998 and 31 December 2011. Two coders independently coded all the reports using a pre-defined framework. Determination of coverage of “standard” and “non-standard” therapies and tests was made by comparing the media print/broadcast date to the date of Federal Drug Administration approval or incorporation into clinical guidelines. Results: Although the term “personalized medicine” appeared in all reports, it was clearly defined only 27% of the time. Stories more frequently reported PM benefits than challenges (96% vs. 48%, *p* < 0.001). Commonly reported benefits included improved treatment (89%), prediction of side effects (30%), disease risk prediction (33%), and lower cost (19%). Commonly reported challenges included high cost (28%), potential for discrimination (29%), and concerns over privacy and regulation (21%). Coverage of inherited DNA testing was more common than coverage of tumor testing (79% vs. 25%, *p* < 0.001). Media reports of standard tests and treatments were common; however, 8% included information about non-standard technologies, such as experimental medications and gene therapy. Conclusion: Confusion about personalized cancer medicine may be exacerbated by media reports that fail to clearly define the term. While most media stories reported on standard tests and treatments, an emphasis on the benefits of PM may lead to unrealistic expectations for cancer genomic care.

## 1. Introduction

With former President Obama’s unveiling of the Precision Medicine Initiative, and the recent Cancer Medicine Moonshot Initiative, interest within the federal government, the medical research community, and the general public in genomic and personalized medicine has been burgeoning [1,2,3]. When defined in scientific literature, personalized medicine refers to a range of concepts, such as the use of biomarkers, genetic and phenotypic information, to personal health behaviors (e.g., nutrition) and preferences [4]. Germline and somatic genomic testing has been incorporated into the care of thousands of cancer patients yearly, and is guideline-endorsed for select patients with malignancies such as non-small cell lung cancer [5], breast cancer [6], chronic myeloid leukemia [7], melanoma [8,9], and colorectal cancer [10]. Given that the promise of cancer genomic medicine is great, but that the ubiquitous implementations of genomic testing and targeted therapies are limited by a myriad of factors, it is imperative that we understand how cancer genomic medicine is being reported in the news media, and whether the news media is helping to set realistic expectations for cancer care.

Prior work has demonstrated the importance of the news media as a source of information for cancer patients [11]. The news media plays a critical role in health education, including playing an important role in disseminating information about emerging technologies, and amplifying the impact of research findings within the medical community [12,13,14]. Research has highlighted prominent links between media exposure and patients’ opinions about the efficacy of therapy, feelings of hope and confusion [15], health behaviors [15,16,17], and health care utilization [17]. Media coverage may also influence government policy, as it did with the provisioning of Herceptin in Canada and Australia [18,19], and it can influence public debate regarding new and controversial therapies [15]. However, media reports are not always balanced and a number of studies have demonstrated an uneven reporting of benefits as compared to risks [20,21], as well as exaggerations of patients’ survival and preliminary research findings [22,23]. In the context of cancer, recent research has highlighted that breast cancer screening coverage often over-represents the benefits over the risks, though there is a trend towards a more balanced discussion [24]. This research also notes that simultaneous presentation of risks and benefits of screening can be interpreted as conflict or controversy regarding recommendations [24]. Other studies that have specifically examined media portrayals of genetic research note relatively accurate and balanced scientific reporting, but also highlight problems with news stories containing incomplete information [24,25,26]. For instance, a recent study with breast cancer patients demonstrated that patients’ knowledge and information-seeking regarding personalized cancer therapy is lacking, and is associated with higher education and income levels [27]. Media can facilitate information dissemination and promote further information-seeking for patients across different educational backgrounds and income levels, and, as such, the accuracy of PM-related representation is critical.

Because the news media can influence health attitudes and behaviors, and because “personalized medicine” is a common phrase used in the lay press, we sought to ascertain whether media coverage of personalized cancer medicine is sensationalized, or whether there is realistic coverage of the benefits of and challenges to personalized cancer medicine in the news media. Sensationalism in journalism entails coverage of exciting, emotionally evocative stories to provoke public interest that is also lacking in important information, or biased in presentation. We operationalized sensationalism as news media coverage of tests or treatments without clear evidence of clinical utility (i.e., lacking information about the unproven nature). Given that prior studies have found unbalanced reporting and an exaggeration of preliminary research findings, we hypothesized that there would be more media coverage of personalized tests and targeted therapies without clear evidence of clinical utility than personalized tests and targeted therapies with demonstrated clinical utility.

## 2. Materials and Methods

### 2.1. Sample Selection

Our sampling frame included news reports and transcripts from the highest-circulation U.S. newspapers, U.S. magazines, journals, the Associated Press, and major broadcast networks accessible through the Lexis-Nexis database. Eligible reports between 1 January 1998 and 31 December 2011, were identified using the terms “cancer” and “personalize*” within one word of “medicine”. Articles were excluded from the analysis if they were duplicates; if they were letters to the editor, websites, events calendars, or news summaries; or if they were not in English. Ethical approval was not required as this was an analysis of media reports and there was no participation by human subjects. The research was performed in accordance with Wiley’s Best Practice Guidelines on Publishing Ethics and performed in an ethical and responsible way with no research misconduct.

### 2.2. Coding Instrument

We developed the coding instrument based on the study aims and literature on media framing and agenda setting [28,29,30,31]. The key domains of the instrument included characteristics of the reports (e.g., title, length, publication/print date), definitions of personalized medicine, benefits and challenges of personalized medicine, whether the Genetic Information Nondiscrimination Act (GINA) legislation was mentioned [32], genomic technologies (e.g., test purpose, type of testing, delivery of testing), types of cancer, types of non-cancer diseases, drugs or targeted therapies, and exemplars of genetic testing and treatment (i.e., personalized stories) (Appendix A). Definitions of personalized medicine were restricted to descriptions in which a verb was likened to the term “personalized medicine”, or clear explanations set aside by punctuation. Any other text describing personalized medicine, or using linked examples was coded as ambiguous. If benefits or challenges of personalized medicine were present, then the specific benefits or challenges were captured.

The coding instrument was refined after coding 50 articles. Stories in the final sample were manually reviewed for relevance. Two coders coded all reports, and coding discrepancies were resolved by a third coder. Genetic tests were categorized as having evidence of clinical utility (“standard of care”) or without clear evidence of clinical utility (“non-standard”) based on the results of a Delphi panel [33]. Therapies were categorized as standard or non-standard based on the U.S. Food and Drug Administration (FDA) approval date.

Characteristics of the media and information provided therein were summarized descriptively. The association between the type of media (print versus broadcast show) and mention of personalized medicine was evaluated by Chi-squared tests or, when small frequency counts were encountered, by Fisher’s exact test. McNemar’s test was used to evaluate whether the media contained more information about benefits versus risks of personalized medicine; whether the media contained more information about germline testing versus somatic/tumor testing; and whether genomic testing delivery was more often ‘clinic’, ‘direct to consumer’, or ‘research’ (with pairwise comparisons with each type of genomic testing delivery). All data were managed using REDCap (Research Electronic Data Capture, Vanderbilt University, Nashville, TN, USA), and statistical analyses were conducted using Stata version 13.1 (StataCorp LP, College Station, TX, USA).

## 3. Results

Our initial search yielded 491 news articles. The final sample after removing duplicates and ineligible publications consisted of 396 reports. Ninety percent of these reports were from print sources and ten percent were from broadcast news organizations. The descriptive content of the articles is summarized in Table 1.

### 3.1. Defining Personalized Medicine

Personalized medicine was clearly defined in 27% of reports, ambiguously defined in 55% of reports and not defined at all in 28% of reports (Table 1). Definitions included words and phrases focusing on genetics, genetic testing or genetic sequencing in 72% of cases, and treatment selection, treatment regimens, or improvements to treatments in 80% of cases. Only 13% of the definitions included text referring to risk prediction and only 6% referred to disease prevention.

### 3.2. Benefits and Challenges of Personalized Medicine

When personalized medicine was defined, the benefits were remarked upon in 96% of reports, and the challenges in 48% of reports. Benefits were mentioned significantly more often than challenges (*p* < 0.001). The prevailing benefit discussed was the role of personalized medicine in improving treatment (89%). Table 2 highlights other commonly cited benefits including the ability to predict the risk of developing disease(s) (33%) and a decrease in side effects (30%). Among the reports discussing the challenges of personalized medicine, risk of discrimination (29%)—especially regarding employment and insurance, increased cost (28%), and privacy concerns (21%) featured most prominently.

### 3.3. Genetic Testing

Genetic testing was referenced in 287 of the 396 reports (82%). Germline testing was referred to in 79% of these reports and somatic/tumor testing in 25% of these reports. Germline testing was mentioned significantly more often than somatic/tumor testing (*p* < 0.001). Pharmacogenetic testing and disease or carrier risk were mentioned in 56% and 43% of these reports, respectively (Figure 1). News reports discussing genetic testing referred specifically to gene sequencing technology in 51% of articles, gene chip/microarray/profiling in 14% of articles, and single nucleotide polymorphisms/genome-wide association study technologies each in 8% of articles (Appendix A). The genetic testing delivery method was presented in a research context (66%) more often than in a clinical (26%) or direct to consumer (10%) context (all pairwise *p* < 0.001).

Seventy of the reports that mentioned genetic testing (24%) included reference to at least one specific genetic test. Eighty-two percent reported on a specific genetic test after the test became standard, and thirty-one percent reported on a specific genetic test, including Oncotype DX and EGFR, before the test became standard (Figure 2 and Appendix A). Pharmacogenetic tests—including UGT1A1 and CYP—and HER2 were mentioned in 24% of these articles, and BRCA1/2 in 19%. Fifteen news stories incorporated genetic testing exemplars.

### 3.4. Targeted Therapies Reported

There were 159 media reports (60%) that included one or more examples of a targeted therapy. Ninety-five percent reported on therapies after FDA approval, and eight percent reported on therapies, primarily gene therapy, before FDA approval (Figure 2 and Appendix A). Sixteen news stories incorporated treatment exemplars.

### 3.5. Types of Cancer and Other Diseases Reported

Breast cancer was the most common cancer reported, referenced in 44% of articles (Figure 3 and Appendix A). The next most common types of cancer indicated were colorectal (18%), leukemia (16%), lung (15%), and prostate (14%). There were 250 media reports (63%) that discussed diseases besides cancer—including cardiac (62%), endocrine (38%), and neurological (33%) diseases.

## 4. Discussion

In a large-scale media content analysis, we found that the term personalized medicine was not clearly defined in the majority of news articles. We also found that there was a greater coverage of benefits, as opposed to risks or challenges, of personalized medicine, but that reporting on particular genetic tests and targeted therapies largely occurred after clinical utility had been established.

Many reports did not define personalized medicine, and the majority of those that did included an ambiguous definition. This lack of clarity may be one factor contributing to the low awareness of personalized medicine among cancer patients and the general public [34,35]; however, this ambiguity perpetuated by the media most likely stems from the term’s indeterminate and overly inclusive use in scientific literature. In a systematic literature review, Schleidgen and colleagues found that the phrase “personalized medicine” was found in the title or abstract of 2457 articles, but only defined 28% of the time [36]. Media reports reviewed in our analysis reflected scientific definitions’ emphasis on the genetic dimension of personalized medicine, with 88% of media definitions including information about genetics. However, the lack of uniformity in the definition of personalized medicine is concerning and increases the risk of miscommunication between regulatory agencies, researchers, clinicians, and patients [37].

Our findings also suggest that there is some heterogeneity in the media regarding the portrayal of different genetic tests and targeted therapies. We found an overstatement of the benefits and an under-reporting of the limitations or risks of personalized medicine. These findings are consistent with prior literature in genetic research and prescription medications, which have shown discrepant reporting between benefits and limitations [20,25,26]. The reported benefits focused principally on personalized medicine’s potential to improve treatments, as cited in 89% of articles discussing benefits, and also commonly mentioned enhancements in risk prediction, and decreasing side effects of therapy, whereas the challenges and risks of personalized medicine focused on discrimination, privacy, and cost. Interestingly, personalized medicine was reported to potentially increase and decrease the cost of care. This discordance may in part spring from the lack of consensus regarding technologies that fall within the definition of personalized medicine, but may also reflect a lack of clarity regarding the costs of implementation and the long term economic effects of integrating personalized medicine into routine clinical care [38,39,40,41]. In addition, despite the challenges in determining the clinical utility of much genomic information and new targeted therapies, the tremendous uncertainties of large-scale genomic testing [42], and the need for patient and provider education in genetics [27,43], these issues were infrequently covered.

Reports of personalized cancer targeted therapies were not sensationalized and did not focus on reporting experimental medications; news coverage of drugs prior to FDA approval was infrequent. Given that reporting on FDA-approved therapies may help to set reasonable expectations for care among the general public, this is an encouraging finding. The primary exception to this trend, however, was gene therapy, which was not FDA-approved during this time but appeared in 8% of articles discussing therapies. While it is not immediately clear why gene therapy was such a distinct exception, it may simply reflect reporters’ discussions of future directions in personalized medicine research, and potentially paradigmatic technologies under development. It may also be a result of the heightened media coverage specific to gene therapy, which prior work suggests may be associated with the emergence of new technologies and highly publicized complications in early gene therapy cases. Furthermore, there is also growing popularity of in vitro fertilization and interest in gene therapy as a “miracle cure” in recent decades [44,45]. This coverage may have been further bolstered by the advances in gene therapy for severe combined immunodeficiency disease [46], and the development of clinical trials for sickle cell disease and hemophilia [47,48].

In contrast, cancer genomic tests were frequently discussed in reports prior to the establishment of clear evidence of clinical utility. This finding is not surprising—most genetic tests do not require FDA approval, and can thus be widely marketed before clinical validity has been established [33,49,50,51]. This activity may be curtailed in the near future however, as the FDA considers overhauling their regulatory structure for genetic tests, moving towards a risk-based approach in the regulation of laboratory developed tests [52,53]. Nonetheless, the reporting of non-standard personalized tests might shape public expectations of treatments and care that are less realistic, and is a signal that we need to continue pushing training and advocacy around responsible scientific reporting in the news media [54].

Finally, despite garnering considerable public attention in recent years, only 10% of news reports on personalized medicine discussed direct-to-consumer (DTC) genetic testing, corroborating recent literature [55]. Reporting on personalized medicine in the context of research or clinical care (66% and 26%, respectively) occurred significantly more often. This difference may be attributed to the increased scrutiny of DTC services in recent years, which has seen criticism on the accuracy and clinical limitations of DTC products [51]. This minimized presence of DTC genetic testing in our results may also be due to a tendency to associate DTC services with “recreational genetics”, such as genetic ancestry testing [56,57], and less so with clinical utility.

Of note, Marcon et al. also investigated news portrayals of PM [55]. In contrast to the current study, they had a broader search for general (i.e., not cancer-specific) PM media across the U.S. and Canada, using the FACTIVA database. Their sampling frame comprised of 774 articles, published on later dates than the current study (1 January 2005–15 March 2016). Similar to our findings, they found that the majority of articles defined PM as based on genetics, that the benefits of PM were commonly discussed (92%), that the efficacy of current or future PM-related treatments was commonly reported (78%), and that reporting on DTC services was low. In contrast, they found that fewer reports discussed the challenges of PM (33% vs. 48%). This may be a result of PM-related concerns being more commonly discussed in reports specifically relevant to cancer, or it may be that media portrayal of PM-related concerns may have decreased in recent years with medical advances. Contrary to our findings, concerns about the risk of genetic discrimination occurred in less than 4% of the articles in Marcon’s study (vs. 29% in the current study). Such discordant findings may be reflective of the Genetic Information Nondiscrimination Act (GINA) coming into effect in 2008 [32], which prohibits genetic discrimination in health insurance and employment. GINA was passed towards the end of our study’s date range, which may account for the prominence of discrimination-related concern in our study. Moreover, the most common concern expressed in Marcon’s study was “limited clinical or health value of genetic information”, as portrayed in 11.6% of the articles. This was not a prominent concern in the current study. Such discrepancy may arise from the current study’s focus on cancer, wherein genetic information often demonstrates higher clinical utility for targeted therapies. Finally, because the current study focuses on cancer, we are able to provide rich details about the specific types of cancer genomic tests and technologies that are in the media, as well as the types of therapies that have been portrayed, and if the press covered them prior to or after they were incorporated into standard clinical practice.

In summary, ambiguity about personalized medicine in the media may contribute to patient confusion and lack of awareness. Reports were not overly sensationalized; however, an emphasis on the benefits of personalized medicine fails to convey to the public the complexity involved in fully integrating genomics into medical care. Additionally, increased coverage is needed for some of the challenges that must be overcome before personalized medicine becomes a reality for most patients. Such challenges include provider and patient education, the uncertainties inherent in large-scale genomic analysis and interpretation, and regulatory and reimbursement policies. Increasing awareness of these barriers may help inform consumers and shape a more nuanced and sophisticated public debate regarding personalized medical care.

## Figures and Tables

**Figure 1 jpm-11-00741-f001:**
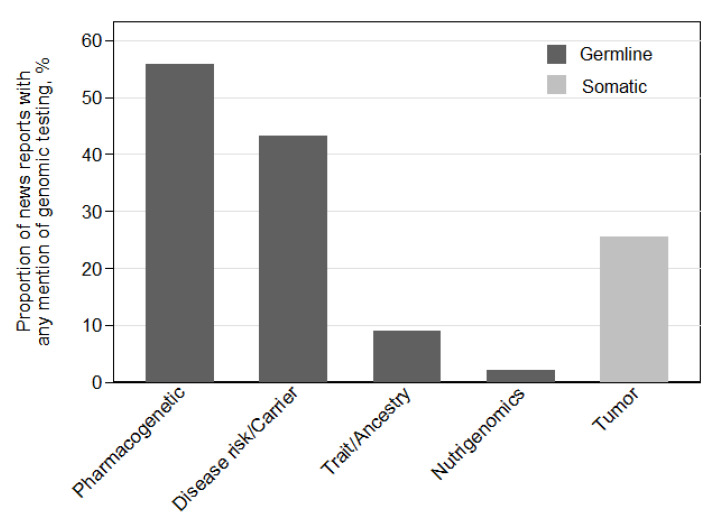
Types of genomic testing reported (*n* = 287).

**Figure 2 jpm-11-00741-f002:**
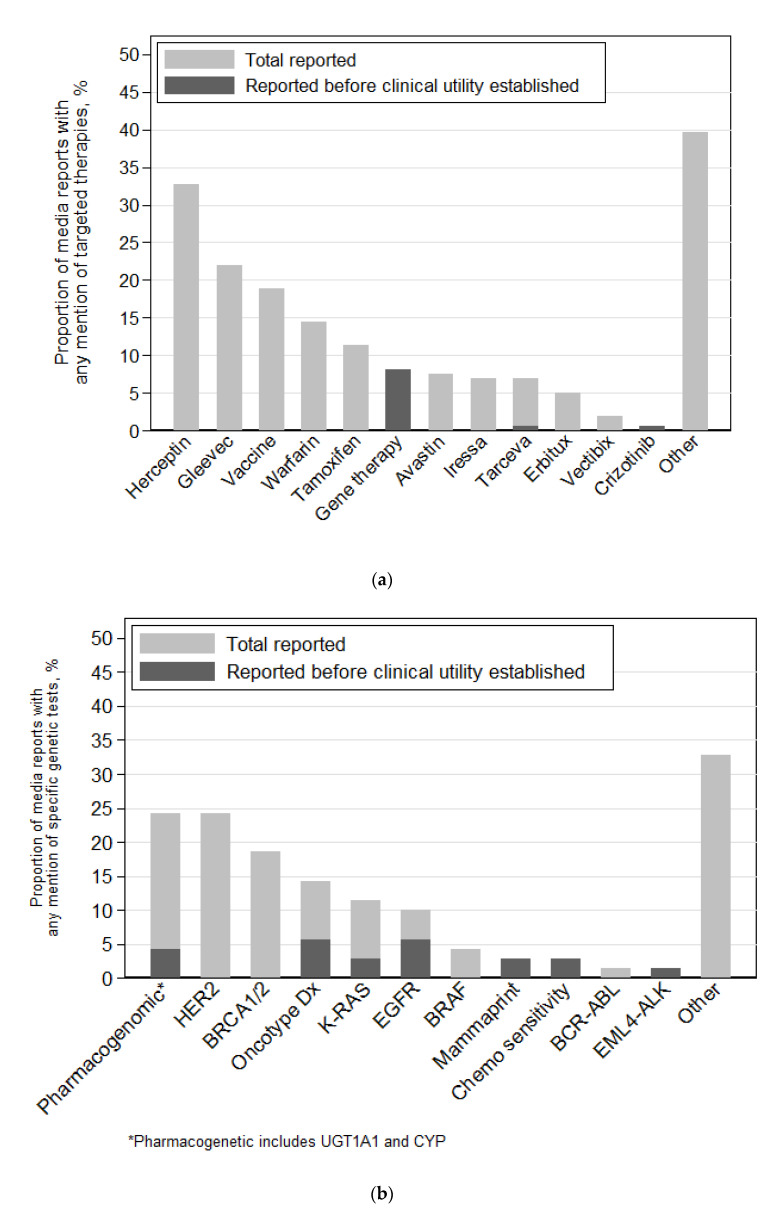
Genetic tests and targeted therapies reported in the media: (**a**) specific genetic tests reported (*n* = 70); (**b**) targeted therapies reported (*n* = 159).

**Figure 3 jpm-11-00741-f003:**
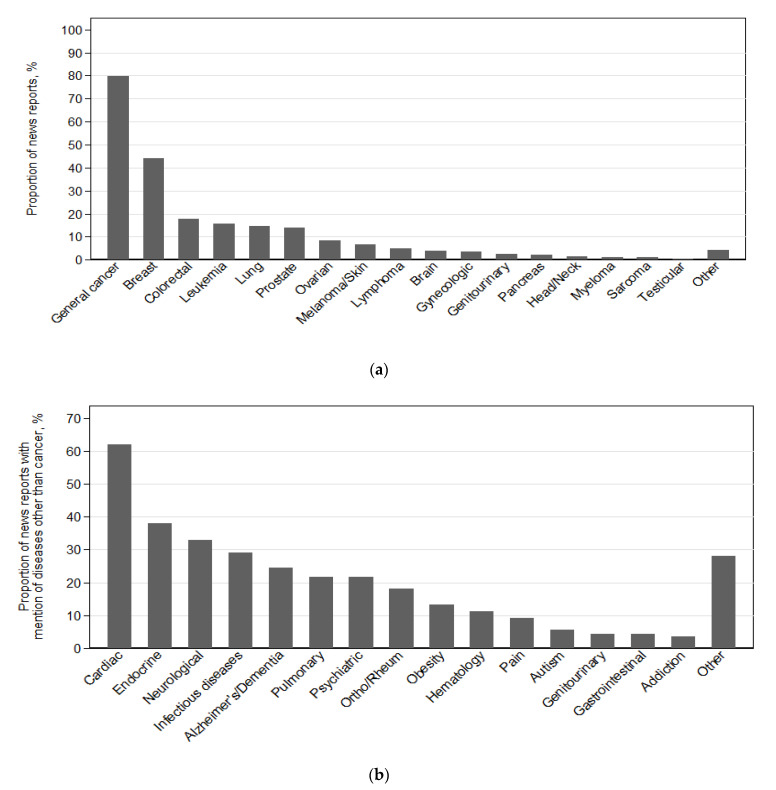
Types of cancer and non-cancer diseases included in media reports about personalized medicine: (**a**) types of cancer reported (*n* = 396); (**b**) other diseases reported (*n* = 250).

**Table 1 jpm-11-00741-t001:** Content of media reports on personalized medicine.

Personalized Medicine	N (%)
Definition of personalized medicine	
No (not defined at all)	110 (28)
Yes (clearly defined)	107 (27)
Ambiguous	179 (45)
Benefits of personalized medicine reported * (*n* = 286)	
No	11 (4)
Yes	275 (96)
Challenges of personalized medicine reported * (*n* = 286)	
No	149 (52)
Yes	137 (48)
**Genetics**	
Genetics mentioned	
No	47 (12)
Yes	349 (88)
Genomic testing mentioned (*n* = 349)	
No	62 (18)
Yes	287 (82)
Specific genetic tests mentioned (*n* = 287)	
No	217 (76)
Yes	70 (24)
Germline testing mentioned * (*n* = 287)	
No	59 (21)
Yes	228 (79)
Somatic/tumor testing mentioned * (*n* = 287)	
No	214 (75)
Yes	73 (25)
Targeted therapies mentioned	
No	237 (60)
Yes	159 (40)
GINA mentioned	
No	384 (97)
Yes	12 (3)
**Care Delivery Method**	
Direct to consumer * (*n* = 287)	
No	257 (90)
Yes	30 (10)
Clinic * (*n* = 287)	
No	212 (74)
Yes	75 (26)
Research * (*n* = 287)	
No	97 (34)
Yes	190 (66)
Exemplars	
None	303 (77)
Genetic testing or treatment exemplars	93 (23)

* *p*-value < 0.001 for reports of benefits vs. challenges, mention of germline vs. somatic/tumor testing, and pairwise comparisons between the three research delivery methods (McNemar’s test).

**Table 2 jpm-11-00741-t002:** Reported benefits and challenges of personalized medicine.

	N (%)
Benefits, subtypes (*n* = 275)	
Improve treatments	244 (89)
Improve outcomes (any)	43 (16)
Improve chance of cure	1 (0.4)
Increase survival	22 (8)
Decrease disease recurrence	6 (2.2)
Improve response rate	4 (1.5)
Predict risk of developing disease	91 (33)
Decrease side effects	83 (30)
Decrease cost	52 (19)
Improve prevention	49 (18)
Beneficial to drug developers	47 (17)
Improve diagnostic capabilities	36 (13)
Improve prognostication	1 (0.4)
Other	41 (15)
Challenges, subtypes (*n* = 137)	
Risk of discrimination (any)	40 (29)
Employment	30 (22)
Insurance	32 (23)
Racial	4 (3)
Increase costs	39 (28)
Concerns over privacy	29 (21)
Regulation	25 (18)
Insurance reimbursement/coverage	22 (16)
Detrimental to drug developers	20 (15)
Ethical	17 (12)
Contribution of genetic vs. environmental factors	11 (8)
Need for education (any)	10 (7)
Providers	7 (5)
Patients	5 (4)
Uncertainty regarding application of data	8 (6)
Exacerbation of disparities	7 (5)
Inadequate cancer care delivery systems	6 (4)
Patent law	6 (4)
Other	67 (49)

## Data Availability

The data presented in this study are available on request from the corresponding author. The data are not publicly available because they were compiled from multiple sources.

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
