# Peer review of "Personalized Cancer Medicine in the Media: Sensationalism or Realistic Reporting?"

_jpm, 2021, doi:10.3390/jpm11080741_

Round 1

Reviewer 1 Report

This paper is well written - it is interesting and concise. My comments primarily relate to a number of areas that I (as a reader) wanted more clarity or substantiation of claims.

Page 1: the topic is personalised medicine (PM) but this is not directly discussed until paragraph 3. Could the authors give a simple definition of PM early on? Is it an umbrella term for the genomic testing etc discussed in paragraph 1?

Page 3: Coding instrument and process was clearly discussed. I was curious as to what software program was used to manage the data?

Page 10 lines 284-286: This information about the scientific definition of personalized medicine would have been useful earlier on in the article.

Page 10 line 310: The authors have written “Reports on personalized cancer targeted therapies were not sensationalized…” – On what basis was this judgement made? If it was through data analysis it would be interesting to clearly show this to the reader.

Pages 10-11 (lines 318-322): This is a very long sentence and could be rewritten for clarity.

Page 11 (lines 326-327): The authors have written “cancer genomic tests were frequently discussed in reports prior to the establishment of clear evidence of clinical utility.” Can this be substantiated through the earlier data analysis? It reads as though the authors can pinpoint the establishment of clinical utility.

Page 12 (line 372): The authors state “Reports were not overly sensationalized” – as with the previous comment on the use of the word “sensationalized”, can the authors explain how they are determining this?

Author Response

Comment: Page 1: the topic is personalised medicine (PM) but this is not directly discussed until paragraph 3. Could the authors give a simple definition of PM early on? Is it an umbrella term for the genomic testing etc discussed in paragraph 1?

RESPONSE: Thank you for this suggestion. We have moved the content from lines 284-286 that defines personalized medicine into the first paragraph.

Comment: Page 3: Coding instrument and process was clearly discussed. I was curious as to what software program was used to manage the data?

RESPONSE: Data were managed using REDCap. We have included this detail at the end of the methods section.

Comment: Page 10 lines 284-286: This information about the scientific definition of personalized medicine would have been useful earlier on in the article.

RESPONSE: Thank you for this suggestion. As noted above, we have moved these lines into the first paragraph of the manuscript (lines 57-59)

Comment: Page 10 line 310: The authors have written “Reports on personalized cancer targeted therapies were not sensationalized…” – On what basis was this judgement made? If it was through data analysis it would be interesting to clearly show this to the reader.

     RESPONSE: The reviewer makes an excellent suggestion. We have included a definition of sensationalism and our operationalization of this construct in lines 97-103. Given that we operationalized sensationalism as coverage of tests without clear evidence of clinical utility, link between our data and sensationalist coverage is now more explicit.

Comment: Pages 10-11 (lines 318-322): This is a very long sentence and could be rewritten for clarity.

RESPONSE: We appreciate this suggestion and have split this sentence into two shorter sentences

Comment: Page 11 (lines 326-327): The authors have written “cancer genomic tests were frequently discussed in reports prior to the establishment of clear evidence of clinical utility.” Can this be substantiated through the earlier data analysis? It reads as though the authors can pinpoint the establishment of clinical utility.

  RESPONSE: We established evidence of clinical utility for tests and therapies in two ways. Therapies were designated as having clinical utility following the date of approval by the US Food and Drug Administration (FDA). Unlike therapies, most genetic and genomic tests do not require FDA approval and can be marketed and sold before evidence of clinical utility has been established. To establish clinical utility for cancer genomic tests, we convened an expert panel in oncology and cancer genetics to categorize each test as having or not having evidence of clinical utility. We used a modified Delphi process, eliciting feedback on the utility of each test, in iterative rounds until the group reached consensus. The consensus determinations on clinical utility from the Delphi panel was also used as the basis for one or our other published studies (Gray, S.W.; Cronin, A.; Bair, E.; Lindeman, N.; Viswanath, V.; Janeway, K.A. Marketing of personalized cancer care on the web: an analysis of Internet websites. Journal of the National Cancer Institute 2015, 107). The Delphi panel process is noted and referenced in line 140.

Comment: Page 12 (line 372): The authors state “Reports were not overly sensationalized” – as with the previous comment on the use of the word “sensationalized”, can the authors explain how they are determining this?

RESPONSE: As noted above, we operationalized sensationalism as reports that covered therapies/tests without evidence of clinical utility. In this analysis, most reports covered tests and therapies that had either FDA approval (therapies) or ample scientific evidence to support clinical utility as determined by our expert Delphi panel (tests).  

Reviewer 2 Report

The research is good research on scientific dissemination through the media, especially in Personalized Cancer Medicine. However, for the paper to be accepted, the authors must make changes.

First, the theoretical framework is not clear and does not delimit the research problem. This is not a journalism journal, but it is important to talk about several concepts: journalism or scientific dissemination (works such as “Making Health Public, How News Coverage Is Remaking Media, Medicine, and Contemporary Life” by Daniel Hallin or “Journalism and scientific dissemination”, By Carolina Moreno). Also if sensationalism is a variable, the term sensationalism should be better defined. What does realistic information mean? It is not an adjective typical of the communication sciences.

The methodology is good and the hypothesis is clear, but in communication studies it is not usual to mix the different media. The article analyzes written and audiovisual media. They must be differentiated or one must be eliminated. The production of information is different. The impact is different. The role of the journalist is different. They must be differentiated. Media selection also needs to be more justified. It is advisable to work with more hypotheses. The question is general and it is better that it becomes a research question. Why is the study until 2011? I do not understand. Justify.

The results are very descriptive. The conclusions are good, but need to be modified based on the above recommendations. The bibliography should be improved.

Author Response

Comment: First, the theoretical framework is not clear and does not delimit the research problem. This is not a journalism journal, but it is important to talk about several concepts: journalism or scientific dissemination (works such as “Making Health Public, How News Coverage Is Remaking Media, Medicine, and Contemporary Life” by Daniel Hallin or “Journalism and scientific dissemination”, By Carolina Moreno). Also if sensationalism is a variable, the term sensationalism should be better defined. What does realistic information mean? It is not an adjective typical of the communication sciences.

RESPONSE: Thank you for suggesting clarification of the framework for this manuscript. The research can best be viewed through the biomedical authority framework in which the news media participates in health education and the dissemination of health messages and information (Charles L. Briggs and Daniel C. Hallin, Making Public Health: how news coverage is remaking media, medicine and contemporary life, 2016, Routledge, Oxon OX, chapter 1). In the context of personalized medicine, it is critical to know whether media reports include coverage of tests and therapies that are known to be of clinical utility and if there are balanced reports on the benefits and challenges of personalized medicine. Personalized medicine is a complex topic, and the news media has a high degree of influence on popular understanding of medical issues.  We have added the role of the media in health education to line 69-72 and a citation to Briggs and Hallin.

In addition, we have included a definition of sensationalism and our operationalization of this construct in lines 97-103. Given that we operationalized sensationalism as coverage of tests without clear evidence of clinical utility, link between our data and sensationalist coverage is now more explicit.

Comment: The methodology is good and the hypothesis is clear, but in communication studies it is not usual to mix the different media. The article analyzes written and audiovisual media. They must be differentiated or one must be eliminated. The production of information is different. The impact is different. The role of the journalist is different. They must be differentiated. Media selection also needs to be more justified. It is advisable to work with more hypotheses. The question is general and it is better that it becomes a research question. Why is the study until 2011? I do not understand. Justify.

RESPONSE: We appreciate the comment that we are conducting a mixed media analysis. In the context of health communication studies, the mixture of written and audiovisual media is common and has been conducted across a ride range of domains including infectious disease and cancer. To list a few examples:

  • Slater et al, News coverage of cancer in the United States: A national sample of newspapers, television, and magazines. Journal of Health Communication, Vol 12, 2008. (media include: newspapers, television, magazines)
  • Berry, SARS wars: an examination of the quantity and construction of health information in the news media, Health Communication, Vol 21, 2007. (media include: print, radio, television and internet)
  • Kelly et al, The HPV vaccine and the media: how has the topic been covered and what are the effects on knowledge about the virus and cervical cancer, Patient Education and Counseling, Volume 77, issue 2, 2009. (media: newspapers, AP wire, television)
  • Stryker et al, A content analysis of news coverage of skin cancer prevention and detection 1972-2003, Arch Dermatol, 2005: 141 (media: radio, television and newspapers)

As is common with many health-related studies, we included a single hypothesis: that there would be more media coverage of personalized tests and targeted therapies without clear evidence of clinical utility than personalized tests and targeted therapies with demonstrated clinical utility. Our hypothesis is noted on lines 104-107. In addition, we agree that stratifying the analyses by media type would be interesting, but unfortunately the funding period for this work has passed and we cannot conduct additional analyses. The study was conducted until 2011 based on the funding period for data collection and analysis.

Comment: The results are very descriptive. The conclusions are good, but need to be modified based on the above recommendations. The bibliography should be improved.

RESPONSE: We have modified the manuscript based on the reviewer comments and have added a reference to Drs. Briggs and Hallin to the bibliography.

Round 2

Reviewer 2 Report

Modifications and responses are good. Nice job.